# Novel RNA-Seq Signatures Post-Methamphetamine and Oxycodone Use in a Model of HIV-Associated Neurocognitive Disorders

**DOI:** 10.3390/v15091948

**Published:** 2023-09-19

**Authors:** Pranavi Athota, Nghi M. Nguyen, Victoria L. Schaal, Sankarasubramanian Jagadesan, Chittibabu Guda, Sowmya V. Yelamanchili, Gurudutt Pendyala

**Affiliations:** 1Department of Anesthesiology, University of Nebraska Medical Center (UNMC), Omaha, NE 68198, USA; pranavi.athota@slu.edu (P.A.); minhnghi.nguyen@unmc.edu (N.M.N.); vicki.schaal@unmc.edu (V.L.S.); syelamanchili@unmc.edu (S.V.Y.); 2Department of Genetics, Cell Biology, and Anatomy, University of Nebraska Medical Center (UNMC), Omaha, NE 68198, USA; s.jagadesan@unmc.edu (S.J.); babu.guda@unmc.edu (C.G.); 3National Strategic Research Institute, Nebraska Medical Center, Omaha, NE 68198, USA; 4Child Health Research Institute, Omaha, NE 68198, USA

**Keywords:** polysubstance use, methamphetamine, oxycodone, HIV, transgenic rat, RNA sequencing, bioinformatics

## Abstract

In the 21st century, the effects of HIV-associated neurocognitive disorders (HAND) have been significantly reduced in individuals due to the development of antiretroviral therapies (ARTs). However, the growing epidemic of polysubstance use (PSU) has led to concern for the effects of PSU on HIV-seropositive individuals. To effectively treat individuals affected by HAND, it is critical to understand the biological mechanisms affected by PSU, including the identification of novel markers. To fill this important knowledge gap, we used an in vivo HIV-1 Transgenic (HIV-1 Tg) animal model to investigate the effects of the combined use of chronic methamphetamine (METH) and oxycodone (oxy). A RNA-Seq analysis on the striatum—a brain region that is primarily targeted by both HIV and drugs of abuse—identified key differentially expressed markers post-METH and oxy exposure. Furthermore, ClueGO analysis and Ingenuity Pathway Analysis (IPA) revealed crucial molecular and biological functions associated with ATP-activated adenosine receptors, neuropeptide hormone activity, and the oxytocin signaling pathway to be altered between the different treatment groups. The current study further reveals the harmful effects of chronic PSU and HIV infection that can subsequently impact neurological outcomes in polysubstance users with HAND.

## 1. Introduction

Polysubstance use (PSU) is a rapidly growing form of substance use disorder (SUD), involving the combined use of more than one type of drug to elicit the user’s desired effect [1]. Commonly, polysubstance users will start off by abusing one drug often enough to be categorized as having SUD. Eventually, the user will develop a tolerance towards it or become tired of the repeated effects, prompting them to explore the combined effects of multiple drugs [2]. Using drugs from different classifications together, such as stimulants (i.e., methamphetamine and heroin) and depressants (i.e., alcohol and opioids), can help individuals feel new and addictive sensations, leading to long-term PSU. This form of PSU is also known as “speedballing”. However, using multiple drugs at once makes it difficult for individuals to track the amount of each drug in their system. Furthermore, it is not possible to predict how each individual’s body will react to the combined usage of more than one drug. This can lead to an increased chance of overdosing on drugs. In 2019, it was reported that one-third of drug overdose deaths in the United States involved the co-use of stimulants and opioids [3]. Many studies have examined the impacts of singular drug use, allowing scientists to develop treatment methods for SUD. Unfortunately, in the context of PSU, these treatments are ineffective due to the lack of research focused primarily on multiple drug use.

Therefore, the present study investigates the combinatorial effects of methamphetamine (METH), a notorious stimulant, and oxycodone (oxy), a commonly prescribed opioid. Mixing drugs with opposing effects, like stimulants that increase activity in the central nervous system (CNS) and depressants that slow down activity in the CNS, aids chronic drug users in dealing with the negative consequences of the drugs associated with individually using them [4]. These drug classifications are known to permeate the blood–brain barrier (BBB) and reach the CNS, encompassing the brain and spine [5]. Thus, chronic drug use can cause permanent changes in an individual’s behaviors and cognitive function.

PSU can lead to increased impulsivity due to delayed processing and poor judgment skills. This puts substance users in danger of contracting incurable, life-altering diseases like human immunodeficiency virus (HIV). While under the unpredictable influence of multiple substances, individuals may partake in high-risk behaviors such as unprotected sex, violence, and suicide [6,7]. Polysubstance users are also more likely to use intravenous injections to absorb drugs into their bloodstream directly. This allows them to feel the effects of the drugs faster but puts them at a higher risk of HIV infection. Intravenous injections can lead to substance users using a needle more than once or sharing a needle with others; both can increase the risk of spreading the virus [8]. HIV can infect the CNS, allowing it to damage neurons and cause inflammation in the brain. Furthermore, HIV introduces viral proteins to the CNS, making the BBB more permeable and susceptible to other harmful infections [9].

Currently, very little is known about the consequences of chronic PSU in HIV-seropositive individuals. This study utilized RNA sequencing and a bioinformatic analysis to identify key regulated genes affected by HIV and PSU. The RNA was isolated from striatal tissue in the brains of HIV-1 Transgenic (HIV-1 Tg) rats. An animal model featuring HIV-1 Tg rats was used to model the harmful effects of PSU on individuals with HIV at the transcriptional level. Previous studies have utilized these HIV-1 Tg animals, making them widely accepted in the realm of neuroHIV [10,11,12]. The HIV-1 Tg rats express viral proteins in their CNS and blood, causing neurocognitive impairments and behavioral changes similar to those seen in HIV-seropositive humans on cART and with drug abuse [10,11,12]. Additionally, we used high throughput ‘omics’ to identify molecular processes and pathways associated with the adverse effects of chronic HIV and PSU. Overall, this study’s findings point to further directions for developing therapeutic targets to improve the health and quality of life of HIV-seropositive individuals with a history of PSU.

## 2. Materials and Methods

### 2.1. Animals

Male and female Fischer 344 wildtype (WT) and F344 Hsd: HIV-1 Transgenic (Tg) rats aged 6–7 months and bred under licensure were used in the study. The design of the animal strain was based on a previous study [13]. Briefly, Tg animals expressed the transgene that consisted of a HIV-1 provirus with a functional deletion of the *gag* and *pol* regions. A total of *n* = 4 (2 females and 2 males) were used in each treatment group. All animals were housed under constant conditions in a 12-h light–dark cycle and had access to food and water ad libitum. All procedures and protocols were approved by the Institutional Animal Care and Use Committee (IACUC) of the University of Nebraska Medical Center (IACUC protocol: 20-104-01-FC; date: 19 August 2021) and conducted in accordance with the National Institutes of Health Guide for the Care and Use of Laboratory Animals.

### 2.2. Polysubstance Administration

To mimic the start of a chronic PSU regimen, we first subjected the animals to a single drug. METH and oxy were purchased from Sigma Aldrich (St. Louis, MO, USA). The drugs were administered via intraperitoneal (i.p.) injections. Both METH and oxy were dissolved in a sterilized saline solution. METH administration was performed in the first seven days with an escalation paradigm (Figure 1). From the 8th day onwards, the animals received a 10 mg/kg METH + 15 mg/kg oxy co-treatment until day 14. The concentration of oxy used in our animal model mimics the high-dose range equivalent in the induction of chronic analgesia in humans [14]. Control animals received isovolumetric saline during the treatment regimen. The treatment regimen lasted 14 days, and animals were anesthetized and sacrificed one hour after the last dose. Then, the brain and blood plasma were removed from the body. The striatum was isolated from the rest of the brain through microdissection. Since the scope of the study is purely focusing on the changes in the brain under the effect of HIV and PSU, only data from the brain are reported.

### 2.3. Total RNA Extraction, Quality Control, Library Preparation, and RNA Sequencing

The total RNA was extracted from the brain striatum of 16 rats from both sexes and different treatment groups using the Direct-Zol RNA kit (Zymo Research, Irvine, CA, USA) following the manufacturer’s protocol. RNA quantity and quality were ensured using BioTek Epoch (BioTek Instruments Inc., Winooski, VT, USA). Approximately, 1.3–3.6 μg of RNA per sample was sent on dry ice to LC Sciences (Houston, TX, USA) for RNA sequencing.

Library preparation was done by trimming fastq format files with the fqtrim tool (https://ccb.jhu.edu/software/fqtrim, access on 20 December 2022) to remove adapters, unknown terminal bases (Ns), and low-quality 3′ regions (Phred score < 30). The trimmed fastq files were processed by FastQC [15]. FastQC: a quality control tool for high-throughput sequence data, available online at http://www.bioinformatics.babraham.ac.uk/projects/fastqc, access on 20 December 2022, for quality control. Rat mRatBN7.2 (https://www.ncbi.nlm.nih.gov/data-hub/genome/GCF015227675.2/, access on 20 December 2022) and HIV-1 GCF_000864765.1 (https://www.ncbi.nlm.nih.gov/assembly/GCF000864765.1, access on 20 December 2022) reference genome and annotation files were downloaded from NCBI RefSeq. The downloaded rat and HIV-1 ref genome and annotation files were combined as one rat_HIV-1 ref genome file and one rat_HIV-1 annotation file. The trimmed fastq files were mapped to rat_HIV-1 by CLC Genomics Workbench 22 (QIAGEN, Aarhus, Denmark) for RNA-Seq analyses.

Once the transcriptome library was prepared, TPM values (Transcripts Per Kilobase Million) were calculated by dividing the read counts by the length of each gene in kilobases. Gene expressions were normalized using read per kilobase (RPK) in R (version 4.1.2), package DESeq2 [16].

### 2.4. Bioinformatic Data Analysis

After normalization, a Student’s *t*-test was done for RNA analysis to identify RNAs showing significant differences between groups (WT vs. HIV, WT-PSU vs. HIV-PSU, and WT vs. WT-PSU). RNAs were filtered using False Discovery Rates (FDR) ≤ 5%, and *p* < 0.05 were considered significant. Cytoscape plug-in ClueGO was used to perform a gene ontology (GO) analysis on the differentially expressed genes (DEGs) [17]. Canonical pathway analysis was performed using Ingenuity Pathway Analysis (IPA) software (Ingenuity^®^ Systems, Redwood City, CA, USA, www.ingenuity.com, accessed on 17 January 2023) by comparing the differentially expressed proteins against known canonical pathways (signaling and metabolic) within the IPA database [18]. The z-score in IPA was calculated as described [18]. An absolute z-score of ≥2 was considered significant. A positive z-score extrapolated the activation state, while a negative z-score predicted the inhibition state of the upstream regulator using the molecule expression or protein phosphorylation patterns of the molecules downstream of an upstream regulator. A −log (*p*-value) cutoff of 1.3 (*p*-value that was greater than or equal to 0.05) was applied, by default, to show only significant canonical pathways.

## 3. Results

### 3.1. HIV and Chronic PSU Leads to Changes in the Striatum Transcriptome

To corroborate the effects of polysubstance use on HIV transcriptome, we subjected striatal RNA from the wildtype (WT), HIV, wildtype with polysubstance use (WT+PSU), and HIV with polysubstance use (HIV+PSU) groups to high-throughput quantitative RNA sequencing. Post normalization, three comparisons were made between the control and experimental groups to examine the differentially expressed genes (DEGs) that resulted from PSU and HIV independently and together. At a criterion of absolute fold-change greater than 2 and FDR adjusted *p*-value of 0.05, WT vs. WT+PSU had a total of 62 DEGs, HIV vs. HIV+PSU had 232 genes, and WT+PSU vs. HIV+PSU had 193 DEGs. Noteworthy, the majority of the DEGs were protein-coding genes, followed by long non-coding RNA. Interestingly, in the HIV vs. HIV+PSU comparison, 0.86% of the ribosomal RNA (rRNA) was differentially expressed, which accounted for 2 out of 232 genes (LOC120099695 and LOC120102105; Appendix A). For the WT+PSU vs. HIV+PSU comparison, one signal recognition particle RNA (SRP_RNA), Rn7sl1, was differentially expressed with the fold-change of +3.079 (Appendix A). Furthermore, the principal component analysis (PCA) revealed good reproducibility of the biological replicates and overall separation between the groups (Figure 2A–C). Figure 2D illustrates all the DEGs in each comparison and the DEGs overlapping between each set of comparisons. A list of the overlapping DEGs can be found in Appendix A.

Since high-throughput ‘omics’ studies generate many potential hits, it is imperative to illustrate them visually. Therefore, we generated heatmaps of the top 50 DEGs for each comparison (Figure 3).

### 3.2. ClueGO and IPA Analysis Demonstrates the Molecular Activities and Pathways Associated with HIV and Chronic PSU

Next, using the bioinformatics tool ClueGO, we analyzed the biological processes (BP) and molecular functions (MF) enriched with these DEGs. In Figure 4A, the most abundant BP associated with DEGs in the WT vs. WT+PSU group was the hormone activity, accounting for nearly two-thirds of the gene ontology (GO) terms (62.67%). Other GO terms associated with WT vs. WT+PSU were related to cellular responses to progesterone, alkaline, or fatty acids, which accounted for 29.33%, 4.0%, and 1.33%, respectively. Similar to the BP of the WT vs. WT+PSU group, the HIV vs. HIV+PSU and WT+PSU vs. HIV+PSU groups also had DEGs that were associated with hormone activity such as a response to corticotropin-releasing hormones (18.18%—HIV vs. HIV+PSU), and neurohypophyseal hormone activity (17.24%—WT+PSU vs. HIV+PSU) (Figure 4B,C). Interestingly, for HIV vs. HIV+PSU and WT+PSU vs. HIV+PSU, there were relatively many DEGs that had similar BP GO terms, such as the regulation of muscle filaments’ sliding speed (18.18% for HIV vs. HIV+PSU and 13.79% for WT+PSU vs. HIV+PSU) and skeletal muscle activity (9.09% for HIV vs. HIV+PSU and 24.14% for WT+PSU vs. HIV+PSU) (Figure 4B,C).

For MF, the top abundant GO terms associated with WT vs. WT+PSU were ATP-activated adenosine receptor activity (14.29%) and neuropeptide activity (14.29%) (Figure 5A); with HIV vs. HIV+PSU were the regulation of ATPase activity (42.86%) and acidic amino acid transmembrane transporter activity (23.81%) (Figure 5B); and with WT+PSU vs. HIV+PSU, those top GO terms were associated with voltage-gated calcium channel activity (18.18%) and neurohypophyseal hormone activity (18.18%) (Figure 5C). In the comparison between WT individuals and WT+PSU, we observed that DEGs were majorly involved in ATP-activated adenosine receptors (e.g., P2ry1) and neuropeptide hormone activity (e.g., Avp and Oxt) (Appendix A). Noticeably, the GO terms in MF between the WT vs. WT+PSU and HIV vs. HIV+PSU comparisons both consisted of RNA polymerase II sequence-specific DNA binding (8.16% in WT vs. WT+PSU and 4.76% in HIV vs. HIV+PSU), while, in the WT vs. WT+PSU and WT+PSU vs. HIV+PSU comparisons, both consisted of protein kinase inhibitor activity (2.04% and 4.55%) and protein homodimerization activity (2.04% and 4.55%). On the other hand, the only GO term that was commonly seen between the HIV vs. HIV+PSU and WT+PSU vs. HIV+PSU comparisons was cytochrome-c oxidase activity (4.76% and 4.55%).

We further investigated potential enriched pathways associated with the DEGs using Ingenuity Pathway Analysis (IPA). As seen in Figure 6A, the pathway associated with the oxytocin signaling pathway was enriched and highly activated in WT vs. WT+PSU. In Figure 6B, the pathways associated with HIV vs. HIV+PSU were highly enriched in several signaling pathways, such as dilated cardiomyopathy (DCM), macrophage alternative activation, and IL-6. In contrast, pulmonary fibrosis idiopathy, G-protein coupled receptors, calcium signaling, ILK, and many other signaling pathways were downregulated. Figure 6C shows that the DCM signaling pathway was highly enriched, similar to Figure 6B. Also, in Figure 6C, many signaling pathways, such as calcium, oxytocin, estrogen receptor, GPCR-mediated nutrient sensing, and opioid signaling pathways, had significant negative z-scores, indicating high deactivation. Notably, in both the HIV vs. HIV+PSU and WT+PSU vs. HIV+PSU comparisons, the genes Tnnc1, Tnnt2, and Actc1 were found to be majorly involved in the calcium signaling and dilated cardiomyopathy (DCM) signaling pathways. Gene-to-disease associations are provided in Appendix A.

## 4. Discussion

Currently, little research has examined the impacts of PSU on the central nervous system (CNS), especially in HIV-1-seropositive individuals. In this study, we utilized an in vivo HIV-1 Transgenic (HIV-1 Tg) rodent model to investigate the effects of the combined use of methamphetamine (METH) and oxycodone (oxy) in the CNS. Through RNA sequencing and bioinformatic tools, we generated a comprehensive transcriptomic landscape of the brain in a rodent model of HAND to discover novel targets and potentially provide new insights into future developing therapeutic strategies for those suffering from HIV and PSU.

Our ClueGO analysis for both BP and MF found very interesting results. First, for the comparison between WT and WT+PSU, we observed that DEGs were majorly involved in ATP-activated adenosine receptors (e.g., P2ry1) and neuropeptide hormone activity (e.g., Avp and Oxt). The P2ry1 gene is a protein-coding gene that codes for G-protein-coupled P2Y receptors. Previous studies about microglia in culture have shown that P2RY1 is responsible for rapid microglial membrane disturbance and whole-cell migration via ATP signaling [19,20]. Importantly, human and animal studies have shown that the chronic use of most drugs of abuse causes a significant increase in peripheral and brain inflammation signals, which trigger microglia and astrocyte activation in the brain. Microglia and/or astrocyte activation has been demonstrated in animal models of every drug studied, including amphetamines, cocaine, ethanol, opioids, cannabinoids, and nicotine [21]. In this study, we identified that the P2ry1 gene is upregulated upon exposure to METH + oxy. This highly suggests that the induction of METH + oxy could trigger P2ry1 as a responding mechanism for potential inflammation. On the other hand, the Avp and Oxt genes are downregulated (Appendix A). The Arginine vasopressin (Avp) gene is encoded for neuropeptide vasopressin, which contributes to balancing the body’s osmosis, balances the blood pressure, maintains sodium homeostasis, and regulates kidney function [22]. Avp is also important in facilitating or promoting aggression [23]. Memory, one of the first Avp brain functions studied, is particularly important in social recognition. Stress-related corticotropin-releasing factor (CRF) and vasopressin (AVP) peptides are strongly associated with METH addiction-related psychostimulant-induced behaviors [24].

Additionally, we found oxytocin (OXT), a 9-amino acid neuropeptide that plays a pivotal role in regulating social bonding, reproduction, and childbirth [25], was downregulated upon exposure to METH + oxy. OXT is synthesized in the hypothalamus and released into the bloodstream by nerve endings in the posterior pituitary. In behavioral endocrinology, mounting evidence has shown that OXT is a key molecule that promotes anxiolytics [26] and enhances multiple aspects of social cognition, such as emotion recognition, social perception, empathetic ability, and trust [27,28,29]. Notably, a study by Uhrig and colleagues [28] found evidence to support that the downregulation of OXT expression and receptors in brain regions involved in social cognition may lead to a dysfunction of oxytocin signaling, which is positively correlated with the development of schizophrenia. Our study specifically showed that the combined use of METH and oxy also reduced oxytocin expression. However, the mechanism of how METH and/or oxy use leads to the reduction of oxytocin synthesis is unclear. Whether using METH and oxy together would have a significant summative effect on the oxytocin level and signaling remains unknown. The rising investigation on the effects of oxytocin treatment on various drug-seeking and drug-induced behaviors has shown the potential of oxytocin in diminishing drug use [30,31,32,33]. However, these studies mostly pertain to single drug addiction, while, in the real world, people can be exposed to multiple drug use simultaneously. Hence, combining the results of our study and the potential therapeutic effect of oxytocin, it could be a promising investigation topic for determining whether oxytocin treatment is a powerful tool to alleviate PSU effects.

Interestingly, in both the HIV vs. HIV+PSU and WT+PSU vs. HIV+PSU comparisons, Tnnc1, Tnnt2, and Actc1 were found to be majorly involved in the calcium signaling and dilated cardiomyopathy (DCM) signaling pathways. Our study found that these three genes were downregulated, suggesting potential dysregulation in the calcium signaling pathway. This pathway significantly contributes to synaptic activity and neurotransmission [34]. Significantly, HIV proteins have also been shown to cause calcium dysregulation in infected neurons, which can induce neuronal death or loss of function [35]. Thus, HIV-seropositive individuals who are chronic substance users may experience significantly more brain damage, despite receiving modern cARTs.

## 5. Conclusions

In summary, this study, for the first time, elucidates the impact of chronic HIV infection along with the effects of the combined use of METH and oxy in a model of HAND. The novel gene signatures, including associated biological processes and molecular functions, could further be developed to inform therapeutic treatments and in mitigating CNS dysfunction in these individuals.

## Figures and Tables

**Figure 1 viruses-15-01948-f001:**
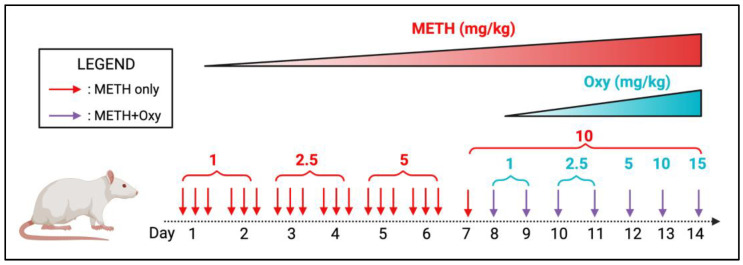
Preclinical rodent model mimicking METH + oxy use. Each arrow represents a single drug dose with the amount of dose (in mg/kg) noted on the top. METH and oxy dosages are written in red and blue, respectively.

**Figure 2 viruses-15-01948-f002:**
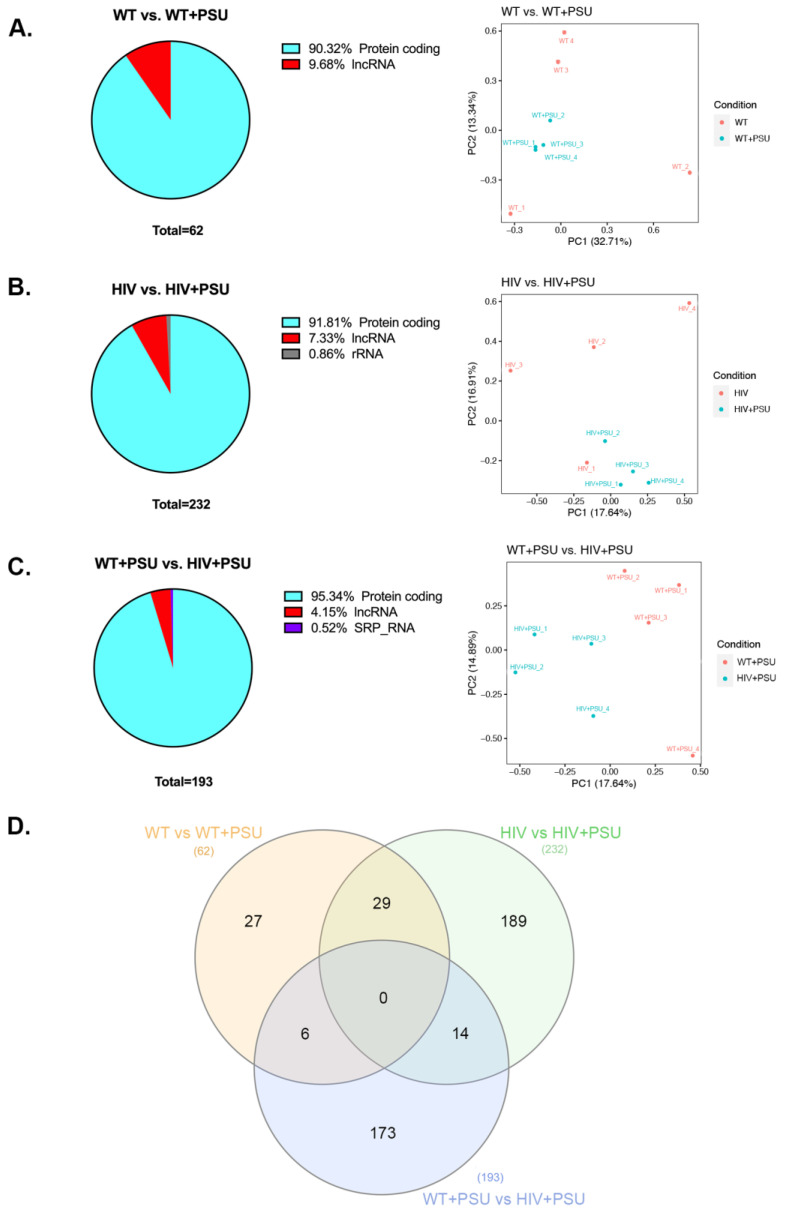
Demographic of DEGs from RNA-Seq data. (**A**–**C**) Percentage of the different types of RNA (those that are differentially expressed) identified by RNA-seq data and the principal component analysis (PCA) between each comparison: (**A**) WT vs. WT+PSU, (**B**) HIV vs. HIV+PSU, and (**C**) WT+PSU vs. HIV+PSU. lncRNA = long non-coding RNA; rRNA = ribosomal RNA; SRP_RNA = signal recognition particle RNA. (**D**) Venn diagram showing total differentially expressed genes found in between the samples (*n* = 4/group).

**Figure 3 viruses-15-01948-f003:**
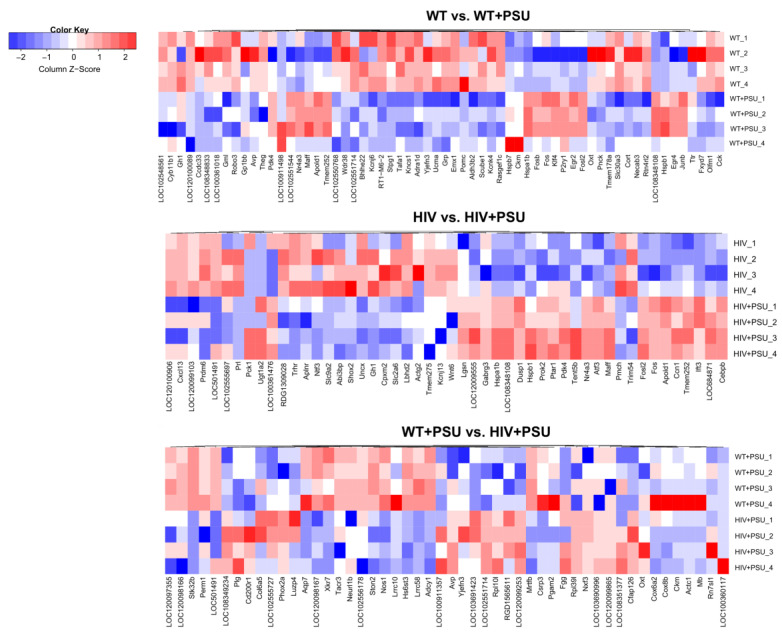
Heatmaps of DEGs. Heatmap visualization of the top 50 differentially regulated genes (DEGs) associated with each comparison: WT vs. WT+PSU, HIV vs. HIV+PSU, and WT+PSU vs. HIV+PSU (*n* = 4/group).

**Figure 4 viruses-15-01948-f004:**
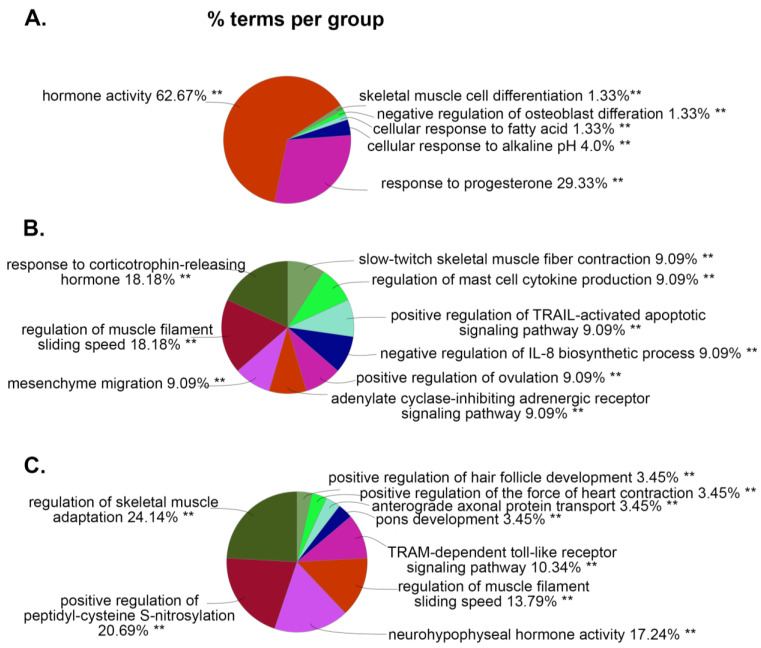
ClueGO analysis of predicted biological processes. Pie charts representing groups of DEGs associated with targeted biological processes in each comparison: (**A**) WT vs. WT+PSU, (**B**) HIV vs. HIV+PSU, and (**C**) WT+PSU vs. HIV+PSU. ** *p* < 0.01.

**Figure 5 viruses-15-01948-f005:**
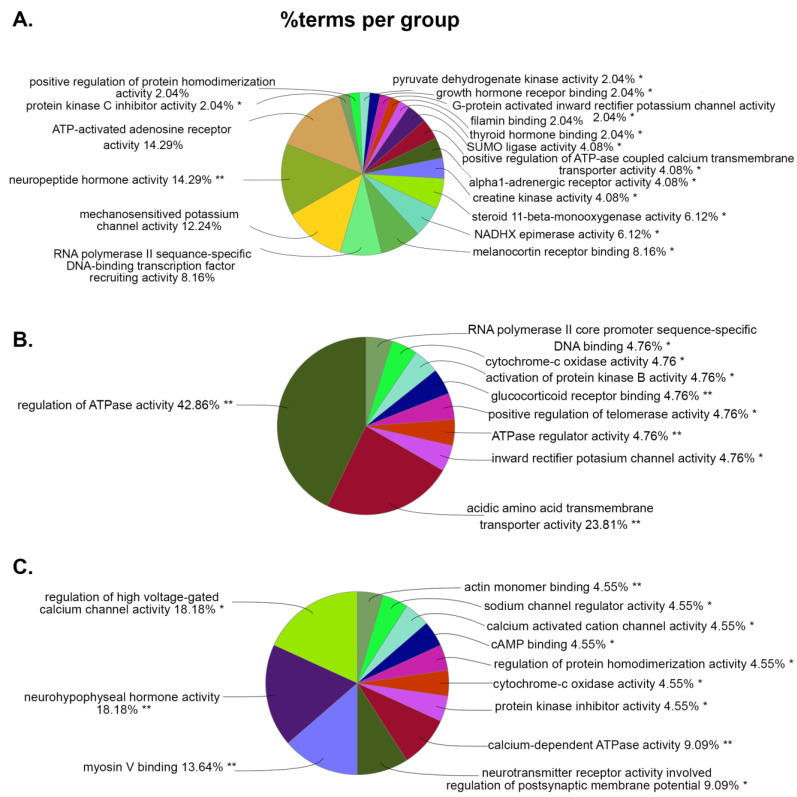
ClueGO analysis of predicted molecular functions. Pie charts representing groups of DEGs associated with the targeted molecular functions in each comparison: (**A**) WT vs. WT+PSU, (**B**) HIV vs. HIV+PSU, and (**C**) WT+PSU vs. HIV+PSU. * *p* < 0.05 and ** *p* < 0.01.

**Figure 6 viruses-15-01948-f006:**
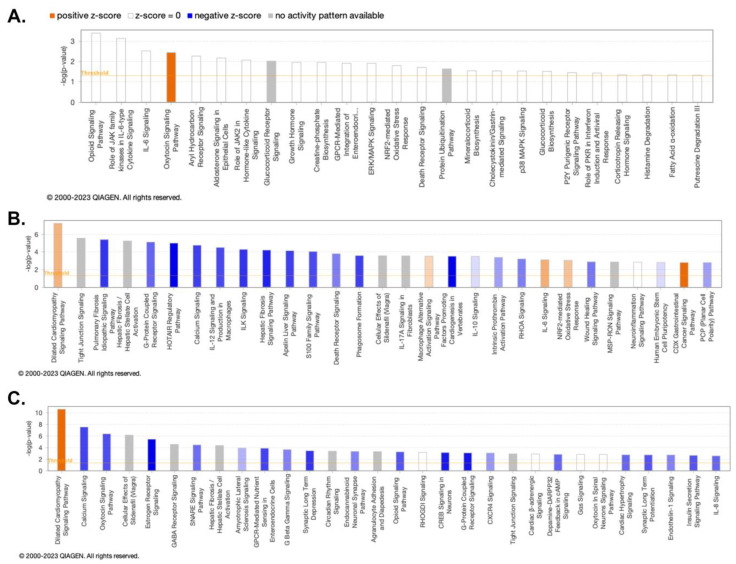
DEGs Ingenuity Pathway Analysis (IPA). Enriched pathways associated with the core pathways were analyzed by IPA, which identified the relationships, mechanisms, functions, and pathways relevant to a dataset, where the orange bar represents a positive z-score of ≥2 (activated pathway) and the blue bar represents a negative z-score ≤ −2 (inhibited pathways). The orange line represents the threshold value of the cutoff *p*-value. By default, IPA applies a −log (*p*-value) cutoff of 1.3, meaning that pathways with a *p*-value equal to or greater than (i.e., less significant than) 0.05 are hidden. (**A**) WT vs. WT+PSU, (**B**) HIV vs. HIV+PSU, and (**C**) WT+PSU vs. HIV+PSU.

## Data Availability

The data discussed in this publication have been deposited in NCBI’s Gene Expression Omnibus (Athota et al. 2023) and are accessible through GEO Series accession number GSE236466 (https://www.ncbi.nlm.nih.gov/geo/query/acc.cgi?acc=GSE236466, access on 20 August 2023).

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
