# Peer review of "Novel RNA-Seq Signatures Post-Methamphetamine and Oxycodone Use in a Model of HIV-Associated Neurocognitive Disorders"

_viruses, 2023, doi:10.3390/v15091948_

Round 1

Reviewer 1 Report

Comments/Questions to reviewers:

1.     Indicate how many rats per treatment group, as well as male:female ratio per group.

2.     Line 94: Indicate how the drug/substance is administered, the volume of the substances given and how often it is given to the rats. Also indicate the source of the drugs and how the drugs are prepared (if they are in solid form).

3.     Line 100: What is “one h”?

4.     Line 101: “…. the brain and blood plasma was harvested”. Details are lacking. Eg:

a)     How are the rats sacrifised?

b)    How was the brain and blood harvested?

c)     How was the brain processed given that the authors specify striatum region of the brain?

d)    What is the purpose of harvesting the blood? Include data from blood analysis.

5.     Line 108: ….extracted from brain cortex….. Should it be “striatum”? Also, do you think 4 rats per group is sufficient for statistical analysis?

6.     Comment on the use of rats at age 6-7 months.

7.     Figure 1: Legend lacking details/information.

a)     What does each arrow represent?

b)    Line 97-98. In the text, explain the rationale in increasing METH and oxycodone dose over time. Why not administer a fixed dose of METH and oxycodone over time?

c)     Comment why lesser injection of oxycodone was given to the rats as opposed to METH.

8.     Line 109-110: Indicate how RNA quantification and quality check was performed.

9.     Section 2.3: Justify the paragraphs.

10.  Comment on why there is no single substance groups (eg transgenic rats receiving only METH and transgenic rats receiving only oxycodone). These groups are crucial to differentiate between single vs polysubstance abuse RNA expression difference, which potentially can lead to targeted treatment for polysubstance abuse, as mentioned in the abstract and introduction.

11.  Comment if there is cognitive function analysis, as well as monitoring the rats physiological conditions (body mass, behaviour) was studied on these rats over the course of the drug administration. If yes, include in the manuscript.

12.  The analysis in the Results section was brief and lacking details. For eg

a)     Section 3.1: Only majority DEGs are highlighted. What about the other genes such as rRNA (that seems to be specific to HIV vs HIV PSU)?

b)    Fig 2A-C:

·      Comment on the low values of PC1 and PC2? Are the values referring to variance?

·      Increase the resolution of the graphs.

·      Comment on the “low” clustering of the rats of the same group. For eg, good clustering seen for WT+PSU & HIV+PSU, but not for WT or HIV alone.

·      Was there comparison done between WT vs HIV? If yes, include the data here. This comparison is important because it provides the fundamental baseline difference between healthy/uninfected rats vs rats expressing HIV-1 viral proteins.

c)     Fig 2D: Comment the overlapping genes between each comparison, as this may be important in identifying common treatment that can be used on HIV+PSU group.

13.  Comment if there is analysis done on quantifying the HIV viral protein and RNA level before and at the end of PSU. There may be useful information by doing such comparison.

14.  There is no narrative/description of analysis for Figure 3. Highlight the major similarity/differences between the gene groups in each comparison.

a)     Is there “phylogenetic” lines above each comparison? If yes, magnify it.

b)    Can the DGE be classified into specific function groups? This will greatly improve clarity of Figure 3.

c)     What does the color code represent?

15.  Section 3.2 title can be improved to indicate what are the key molecular activities and pathways associated with HIV and Chronic PSU.

16.  Figure 4 & 5: Indicate what statistical test was used and what the ** means. Are there non-significant DEG associated with targeted biological processes/molecular function in each comparison?

17.  Figure 6:

A)    Add a scale to indicate the z-scores because there are different shades of orange and blue in the graphs.

B)    What is the threshold line represent? Explain that in the figure legend/main text.

C)    How are the z-score calculated? Explain that in the Methods section.

18.  Line 220: Discussion. There are parts in the discussion that belongs to the Result section. Eg 249-251, 260-261, 274-276 and other relevant parts within Discussion. The Results and Discussion section requires improvement. More importantly, the authors can use the current data to comment on what are the potential treatments for polysubstance abuse in the discussion (both with and without HIV).

19.  In the Result section, can the authors comment if there are differences in the RNA expression/DEG between male and female rats (WT or HIV Tg), as this may impact the hormone responses (Figure 4).

20.  In the abstract, Line 22, authors can specify what are the key markers due to post-METH and oxycodone exposure.

1.     Drugs and substances are used throughout the manuscript. It would be good to use one for consistency.

2.     Abusers or users?

3.     Line 78. Should be “cART” because CART is commonly known as Chimeric Antigen Receptor therapy.

4.     Line 73: HIV-1 Tg or HIV Tg?

Author Response

  1. Indicate how many rats per treatment group, as well as male:female ratio per group.

Thank you for your critique, we have incorporated the information related to male:female in line 95 in the revised manuscript.

  1. Line 94: Indicate how the drug/substance is administered, the volume of the substances given and how often it is given to the rats. Also indicate the source of the drugs and how the drugs are prepared (if they are in solid form).

We have incorporated the information about how the drugs were prepared and administered in lines 109-111. We have also made the Figure 1 legend more detailed and clearer.

  1. Line 100: What is “one h”?

We have clarified this info in the manuscript in line 117.

  1. Line 101: “…. the brain and blood plasma were harvested”. Details are lacking. Eg:
  2. a)     How are the rats sacrificed?
  3. b)    How was the brain and blood harvested?
  4. c)     How was the brain processed given that the authors specify striatum region of the brain?
  5. d)    What is the purpose of harvesting the blood? Include data from blood analysis.

All these concerns have been addressed in the manuscript (lines 116-120). Regarding the information and data about blood analysis, our focus is identifying the markers related to the HIV+PSU, and our tissue focus is the brain. Hence, the blood analysis is unavailable now, and it’ll be used for prospects of our study.

  1. Line 108: ….extracted from brain cortex….. Should it be “striatum”? Also, do you think 4 rats per group is sufficient for statistical analysis?

We apologize for this oversight and it should read striatum, and this change has been incorporated in the manuscript in line 118. Regarding the sample size for RNA-seq, while a larger sample size is generally preferred, we carefully considered practical constraints, and we determined that 4 rats per group were adequate to provide sufficient statistical power to detect meaningful differences. Due to the specificity of high throughput studies and the costs associated we them, we believe our approach is appropriate, considering available resources and ethical considerations.

  1. Comment on the use of rats at age 6-7 months.

Rats aged 6-7 months are representative of middle-aged adults and notably mimic HIV+ individuals with chronic HIV infection including significant changes in the CNS milieu and thus were employed in this study.

  1. Figure 1: Legend lacking details/information.
  2. a)     What does each arrow represent?
  3. b)    Line 97-98. In the text, explain the rationale in increasing METH and oxycodone dose over time. Why not administer a fixed dose of METH and oxycodone over time?
  4. c)     Comment why lesser injection of oxycodone was given to the rats as opposed to METH.

Thank you for your critiques. We added the additional information for the arrow regards to Figure 1. The red arrow on the legend represents the METH-only dosage, and each arrow represents a single drug dose. On day 1, there are 3 arrows, which means that 3 doses of the drug were administered. Similarly, our purple arrow represents METH+oxy combined dosage.

Regarding to the escalation dosage of METH and oxycodone dosage over time instead of a fixed dose, our rationale was that substance users typically build a tolerance to the drugs, so they are likely to start increasing the dose to keep experiencing the same effects.

  1. Line 109-110: Indicate how RNA quantification and quality check were performed.

      We have incorporated the information into the text accordingly.

  1. Section 2.3: Justify the paragraphs.

These paragraphs explain in detail the methods used for total RNA extraction, quality control, library preparation, and RNA-Sequencing.

  1. Comment on why there is no single substance groups (eg transgenic rats receiving only METH and transgenic rats receiving only oxycodone). These groups are crucial to differentiate between single vs polysubstance abuse RNA expression difference, which potentially can lead to targeted treatment for polysubstance abuse, as mentioned in the abstract and introduction.

Thank you for your insightful comment. We appreciate your attention to detail regarding the absence of single substance groups in our study involving transgenic rats. The reason for not including these single substance groups, such as rats receiving only METH or only oxycodone, was a cognizant experimental design choice based on the focus and objectives of our research. In the abstract and introduction, we highlighted our primary interest in exploring RNA expression differences in polysubstance abuse scenarios. Our goal was to investigate the unique molecular changes that occur when multiple substances are used in combination, as this situation is prevalent in real-life polysubstance abuse cases. By comparing the RNA expression profiles in transgenic rats exposed to multiple substances, we aimed to identify specific gene expression patterns associated with polysubstance abuse. These findings have significant implications for developing targeted treatment strategies tailored to individuals with complex substance use disorders involving multiple drugs. While the inclusion of single substance groups could offer valuable insights into the individual effects of METH and oxycodone, doing so would have diverted the focus of our study away from the specific RNA expression differences in polysubstance abuse. Considering the scope and objectives of our research, we opted to prioritize the investigation of gene expression changes arising from the concurrent use of multiple substances. We acknowledge that exploring the individual effects of METH and oxycodone would be an important avenue for future research, and we appreciate your suggestion in this regard. However, our current study aimed to address the pressing need for understanding the molecular underpinnings of polysubstance abuse to advance the development of more effective targeted treatments.

  1. Comment if there is cognitive function analysis, as well as monitoring the rats physiological conditions (body mass, behavior) was studied on these rats over the course of the drug administration. If yes, include in the manuscript.

      Thank you for your valuable feedback and inquiry. We appreciate your interest in the cognitive function analysis and physiological conditions of the rats during the drug administration period. As of now, we do not have specific behavioral data related to the rats' cognitive function during the drug administration period. Since our study primarily focused on RNA-seq data analysis, we did not conduct an in-depth investigation of cognitive impairments or behavioral changes in the rats. However, we agree with your suggestion that exploring cognitive function and behavior in depth would be a valuable aspect to consider for future research. The inclusion of behavioral data could significantly enhance the comprehensiveness of our study and provide a more comprehensive understanding of the effects of drug administration on the rats and. However, we currently lack animals in this age range in our current inventory. Nevertheless, this is an excellent suggestion and will further bolster gaining more comprehensive insights into the effects of polysubstance use on both the molecular and behavioral levels.

  1. The analysis in the Results section was brief and lacking details. For eg
  2. a)     Section 3.1: Only majority DEGs are highlighted. What about the other genes such as rRNA (that seems to be specific to HIV vs HIV PSU)?
  3. b)    Fig 2A-C:
  • Comment on the low values of PC1 and PC2? Are the values referring to variance?
  • Increase the resolution of the graphs.
  • Comment on the “low” clustering of the rats of the same group. For eg, good clustering is seen for WT+PSU & HIV+PSU, but not for WT or HIV alone.
  • Was there comparison done between WT vs HIV? If yes, include the data here. This comparison is important because it provides the fundamental baseline difference between healthy/uninfected rats vs rats expressing HIV-1 viral proteins.
  1. c)     Fig 2D: Comment on the overlapping genes between each comparison, as this may be important in identifying common treatment that can be used in HIV+PSU group.

We have incorporated comments about the unique expression of rRNA and SRP_RNA in the comparisons involving the HIV+PSU group in lines 186-190. Regarding the resolution on the graph, we acknowledge the limitation on the resolution. However, we have tried our best not to compromise the overall size of the figure. For the comparison between WT and HIV, we excluded this comparison in our study for 2 main reasons. First, there is plentiful data available regarding only WT and HIV in rodent transgenic models, as well as other species. Second, the scope of our study is focusing on the effect of PSU on HIV. Altogether, we decided not to display the comparison done between WT and HIV. Regarding the overlapping genes, we have updated this list in Table S1.

  1. Comment if there is analysis done on quantifying the HIV viral protein and RNA level before and at the end of PSU. There may be useful information by doing such comparison.

Thank you for your insightful comment. We agree that analyzing changes in HIV viral protein and RNA levels before and after the polysubstance use intervention could provide valuable insights into the impact of PSU on viral dynamics. Unfortunately, due to the limited availability of samples and resource constraints, we were unable to perform this specific analysis in the current study. As you correctly pointed out, such data would indeed be valuable for understanding the effects of PSU on HIV viral replication. We acknowledge this limitation and understand its significance in shedding light on the potential impact of PSU on HIV-seropositive individuals.

  1. There is no narrative/description of analysis for Figure 3. Highlight the major similarity/differences between the gene groups in each comparison.
  2. a)     Is there “phylogenetic” lines above each comparison? If yes, magnify it.
  3. b)    Can the DGE be classified into specific function groups? This will greatly improve clarity of Figure 3.
  4. c)     What does the color code represent?

For our heatmaps, as we indicated in both results and figure legends, it’s a representation of the top 50 differentially expressed genes (DEGs) associated with each comparison. We have tried our best to have the photo in the right resolution without compensating the text size for each of the genes. The purpose of Figure 3 is to illustrate the most abundant DEGs that appeared in each comparison. Since some DEGs have multiple functions, it isn’t possible to classify them into a specific functional group.

For the heatmap, a Z-score normalization is performed on the normalized read counts across samples for each gene. Z-scores are computed on a gene-by-gene (row-by-row) basis by subtracting the mean and then dividing by the standard deviation. The computed Z score is then used to plot the heatmap.

Genes with dark red are up-regulated, and blue are down-regulated. Since the rows (genes) are Z-Score scaled, the colors represent a single gene’s varying expression across the samples.

  1. Section 3.2 title can be improved to indicate what are the key molecular activities and pathways associated with HIV and Chronic PSU.

      Thank you for your suggestion. However, we decided to keep our current title because we are not focusing on a single particular pathway.

  1. Figure 4 & 5: Indicate what statistical test was used and what the ** means. Are there non-significant DEG associated with targeted biological processes/molecular function in each comparison?

We included the meaning of ** in the figure legend. For each comparison, only the significant DEGs are included for ClueGO analysis since there are too many hits generated.

  1. Figure 6:
  2. A)    Add a scale to indicate the z-scores because there are different shades of orange and blue in the graphs.
  3. B)    What is the threshold line represent? Explain that in the figure legend/main text.
  4. C)    How are the z-score calculated? Explain that in the Methods section.

Thank you very much for your concerns. Since we submitted the data to Qiagen software, their own algorithm calculated the z-score as well as the default threshold. For the IPA z-score it is calculated by the IPA z-score algorithm that Qiagen develops. The Activation z-score predicts the activation state of the upstream regulator using the molecule expression or protein phosphorylation patterns of the molecules downstream of an upstream regulator. An absolute z-score of ≥ 2 is considered significant. An upstream regulator is: Activated if the z-score is ≥ 2 (represented in an orange bar) and inhibited if the z-score ≤ ‑2 (represented in a blue bar). By default, IPA applies a -log (p-value) cutoff of 1.3, meaning that pathways with a p-value equal to or greater than (i.e., less significant than) 0.05 are hidden.

  1. Line 220: Discussion. There are parts in the discussion that belongs to the Result section. Eg 249-251, 260-261, 274-276 and other relevant parts within Discussion. The Results and Discussion section requires improvement. More importantly, the authors can use the current data to comment on what are the potential treatments for polysubstance abuse in the discussion (both with and without HIV).

These changes have been incorporated.

  1. In the Result section, can the authors comment if there are differences in the RNA expression/DEG between male and female rats (WT or HIV Tg), as this may impact the hormone responses (Figure 4).

Great point but this is outside of the scope of our current study which does not focus on investigating sex differences.

  1. In the abstract, Line 22, authors can specify what are the key markers due to post-METH and oxycodone exposure.

Thank you for the suggestion. Unfortunately, since we needed to finalize the abstract before depositing the RNA-seq data, we are no longer able to make changes to the abstract. The key markers are specified in the Results section (lines 240-243, 265-268) and further explained in the Discussion section.

Comments on the Quality of English Language

  1. Drugs and substances are used throughout the manuscript. It would be good to use one for consistency.
  2. Abusers or users?
  3. Line 78. Should be “cART” because CART is commonly known as Chimeric Antigen Receptor therapy.
  4. Line 73: HIV-1 Tg or HIV Tg?

Thank you for all of the comments, we have edited and changed our text based on your suggestions and kept everything consistent.

Reviewer 2 Report

1. Lack of detailed information on HIV 1-tg mice

2. Lack of validation experiments

3. Lack of experiments in behavioral or pathological aspects

4.Streatum should not be an extension of Cortex, and the author needs to explain whether RNA originates from Cortex or Streatum

Author Response

  1. Lack of detailed information on HIV 1-tg mice

Thank you for your comment. We have addressed the HIV Tg rat model in more detail, provided in lines 91-100 of the manuscript.

  1. Lack of validation experiments

Thank you for your comment. We acknowledge this limitation. The nature of RNA-seq and the limited amount of Striatal tissue isolated was a limitation for us to further validate the DEGs identified in this current study and are apt for future studies, including probing the downstream mechanisms.

  1. Lack of experiments in behavioral or pathological aspects

Thank you for your comment. We acknowledge this limitation. However, our focus for this study is to identify the biomarkers that associated with HIV+PSU. This study is purely an exploration/discovery-based study. Therefore, behavioral and pathological aspects is not in the scope of our focus at the moment.

4. Striatum should not be an extension of Cortex, and the author needs to explain whether RNA originates from Cortex or Striatum.

Thank you for your comment. It was a mistake on our end. We have addressed this issue in line 118.

Round 2

Reviewer 2 Report

The authors still didn't validate the DEGs identified in this study.

Author Response

As mentioned in our previous response, due to the constraints of RNA-seq and the limited Striatal tissue available on our end, we were unable to validate the identified DEGs in this study. Please note that striatal tissue in rats is relatively small, and we have already used all our samples for isolating enough RNA for sequencing. We acknowledge this limitation and view it as a potential avenue for future research, including exploring downstream mechanisms. This limitation has also been acknowledged by the academic editor in his comments and the paper is now acceptable. We thank you for your wonderful insights during the review process and greatly appreciate it.